# Optimization of the Sustainable Distribution Supply Chain Using the Lean Value Stream Mapping 4.0 Tool: A Case Study of the Automotive Wiring Industry

**Yousra El Kihel [1],\*, Ali El Kihel [2] and Soufiane Embarki [3]**

1    IUT of Bordeaux-Bastide, Department MLT, University of Bordeaux, 33072 Bordeaux, France
2    Faculty of Sciences and Technologies, Hassan 1st University, Settat 26000, Morocco
3    Industrial Engineering Laboratory, National School of Applied Sciences, Oujda 60000, Morocco
\*    Correspondence: elkihel.yosra@gmail.com; Tel.: +33-66-696-9023

**Abstract:** The transformation to Supply Chain (SC) 4.0 promises new opportunities for companies to gain competitiveness. The Lean Value Stream Mapping (VSM) tool allows the supervision of all the processes of the entire SC, from which we can identify the different types of waste that hinder the competitiveness of the SC. Following the existing problems detected with the help of a diagnostic, we will propose a new process design by integrating 4.0 technologies to modernize the company. For our case study, we treat the multinational SC of Automotive Wiring Equipment Morocco, where we will focus on the downstream part of the SC composed of the warehouse and the different stages of road and sea transport until the final delivery in Austria. Then, we will consider the opportunities offered by each country in terms of logistics competitiveness. In this research work, we will show how Lean VSM4.0 will contribute to sustainable development by integrating the three pillars economic, environmental, and social. With the Lean VSM 4.0 tool, all logistic processes will be visualized, from which improvements could be obtained, especially the optimization of the lead-time, the cost, the energy consumed, and the follow-up of the products during the whole SC while reducing accidents.

**Keywords:** VSM; durability; logistics 4.0; methodology; automobile; wiring

## 1. Introduction

Currently, there is increased competition between companies in the automotive industry. The automotive equipment manufacturers, cabling, and others are characterized by very high competitiveness and diversity of products and distribution on an international scale. Morocco has become the first construction hub on the African continent, ahead of South Africa and Egypt, with distribution on an international scale [1]. Therefore, it is necessary for each company to adapt to new technological changes to increase production and find solutions for improving the flexibility of its distribution system by delivering to customers at a low cost with a deadline and optimal quality [2].

Enterprise 4.0 refers to a new generation of connected and intelligent factories. It is a digital transformation characterized mainly by intelligent automation and the integration of new technologies into the enterprise value chain, giving life to an interconnected enterprise 4.0 where machines, products, and people interact [3]. The digitization of the SC, according to Industry 4.0, therefore, promises new opportunities to develop more efficient supply, production, and distribution lines, while supervising the entire supply line.

Digital twins play a crucial role in supporting and improving industrial manufacturing processes [4,5]. Traditional value stream mapping (VSM) methods are suitable for planning-based production process reengineering. Some studies introduce the methodological contribution to integrating digital twins and VSM for SMEs. This approach is based on a simulation to improve quantitative analysis in the production process by reengineering tasks and logistics [6] and provides a framework based on the digital twin to use physical

product data, virtual product data, and connected data that link physical and virtual products for product design, manufacturing, and service. It is possible to design digitized views of the different actors in the chain and simulate their reactions. The digital twins involve partners, suppliers, customers, etc., and all actors in the company's supply chain.

VSM 4.0 is one of the effective tools to analyze, optimize the process, and eliminate waste [7].

The VSM 4.0 method, which is easy to implement, makes it possible to identify, with the help of a diagnostic, classic waste as well as information wastes and to design lean value flows in terms of products and, above all, of information flows [8]. Thanks to the VSM 4.0 application, it is now possible to perform a value stream analysis, to create a new value stream design based on new digital technologies [9].

Our problem involves studying, analyzing, and optimizing the efficiency and effectiveness of a logistic chain of the multinational Automotive Equipment Manufacturer EAKM located in Kenitra, which knows is in hard competition with other companies. We will first focus on the downstream part of the SC which concerns the shipping area, preparation of orders, and loading into trucks in the factory. Then, we will investigate the different stages of road and sea transport from Kenitra to Vienna according to the following route: Morocco, Spain, France, Italy until delivery to the EAVA warehouse in Austria. Following the different types of existing waste identified, we will propose modern technologies to be integrated into the logistic line as solutions for a better performance of the whole SC. The SC functions will be connected by 4.0 technologies, with a visualization of all the tasks with the VSM 4.0 tool. The results obtained are very satisfactory, particularly the collection of data, the optimization of the functions of the operators in Morocco and Vienna, automation of several workstations, tracking of products throughout the supply chain, and optimization of transport time between the two countries.

For the working methodology, we present a literature review on VSM 4.0, its usefulness in Lean Logistics, and the different modern 4.0 technologies compatible with VSM with their advantages. Next, we perform a VSM mapping of the current state and define the different tasks of the process. Finally, by analyzing the current state, we will propose improvements for establishing the future state of our VSM 4.0. The results obtained are also part of a sustainable development framework through improving the economic, social, and environmental components.

## 2. Literature Review

### 2.1. Lean Production System

In manufacturing systems, research is focused on reducing process variability, increasing productivity, and developing new processing methods to reduce costs, implement predictive models, increase quality, and adopt intelligent automation and robotics solutions by adopting lean manufacturing [10].

The Lean concept is a solution to reduce the complexity of automated production systems. One of the latest developments of the adaptive production system is Lean Automation, which allows the system to adapt to future market requirements by achieving high flexibility and short-flow information. This new theory offers the perspective of integrating the Industry 4.0 concept into a Lean production system [11].

According to Lean theory, there is a set of tools and applications through which waste is identified and eliminated or reduced. Also, a set of Industry 4.0 fundamental technologies has been identified through which losses are identified, managed, or eliminated [12].

In this research work, we are interested in the lean VSM tool and how the new technologies of Industry 4.0 can contribute to it. In addition, VSM 4.0 eliminates redundant steps in information logistics along the value chain.

In the following paragraph, we conduct a literature review on SCOPUS to further our research on this topic.

## 2.2. Research in VSM 4.0 Published between 2016 and 2021

We used Scopus, a scientific research platform, to determine the number of publications on VSM 4.0 between the years 2016 and 2021 (Figure 1).

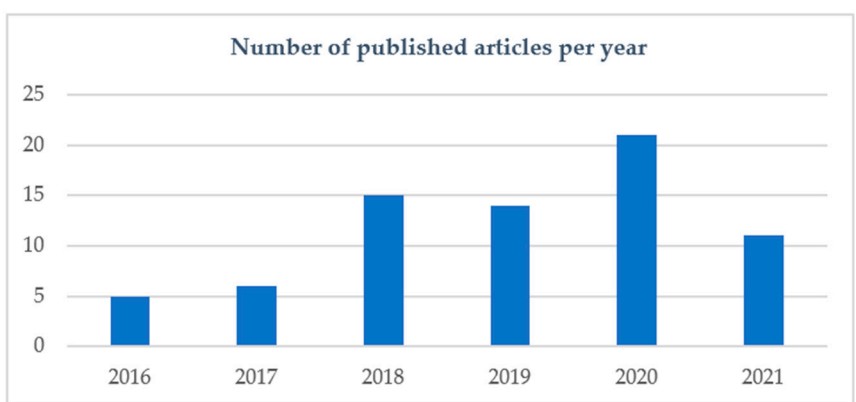

**Figure 1.** Number of publications from 2016 to 2021 on VSM 4.0.

According to the results of the bibliographic research, the total number of articles remains low, showing that the VSM 4.0 is a new and very interesting topic, which researchers and industrialists are developing more and more.

### 2.2.1. Publications of the Theme in Relation to New Technologies between 2016 and 2021

We conducted a more in-depth bibliographic study on the SCOPUS search engine only for scientific articles that were published between the years (2016–2021).

The objective of this study was to show the relationship between 4.0 technologies and the VSM tool used in the literature or in the industry. The results obtained are presented in Figure 2. We noticed that IoT technologies and Big Data are the most frequently cited. In addition, the rest of the technologies, such as cloud computing, robots, artificial intelligence, etc., remain useful for the VSM 4.0. These results can be explained by the large amount of information to manage, the physical flow of products, the number of stakeholders, and the SC.

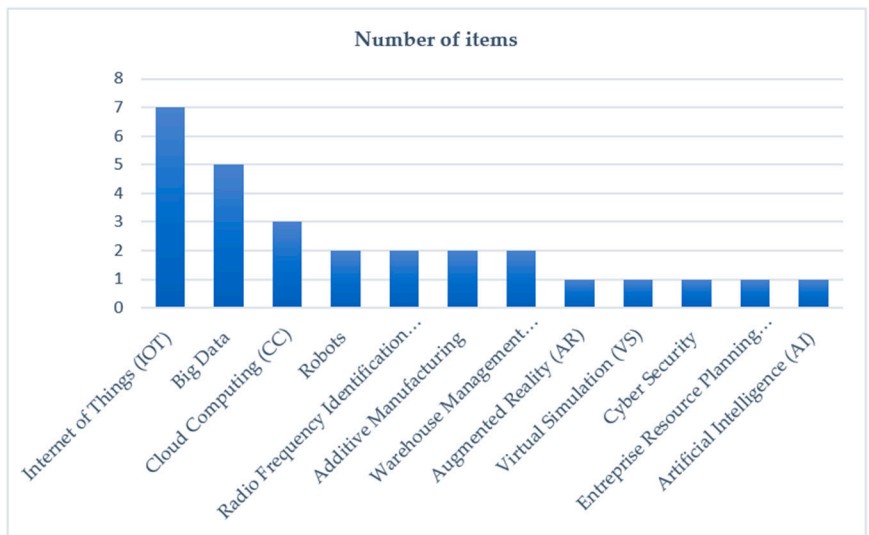

**Figure 2.** Number of publications of 4.0 technologies used in VSM4.0.

### 2.2.2. Publications sur VSM Classique et VSM 4.0

Table 1 presents the research work (17 articles) on the VSM tool with the integration of new technologies for the last few years and, particularly, the benefits it can offer to logistics activities.

**Table 1.** Literature review from 2015 to 2022.

| N° | Year | Title | Authors | Type | Contenent |
|---|---|---|---|---|---|
| 1 | 2022 | Extending the lean value stream mapping to the context of Industry 4.0: An agent-based technology approach [12] | Ferreira, Armellini, F., Santa-Eulalia, Thomasset-Laperrière | Article | Integrating VSM with hybrid simulation (HS) and extending its scope to the Industry 4.0 context |
| 2 | | A cyber-physical system architecture based on lean principles for managing industry 4.0 setups [13] | Nounou, Jaber, H., Aydin, | Article | Smart VSM 4.0 to improve material and information flow and autonomous decision making via IOT |
| 3 | | An Integrated Value Stream Mapping and Simulation Approach for a Production Line: A Turkish Automotive Industry Case [14] | Aksar, O., Elgun, D., Beldek, T., Konyalıoğlu, A.K., Camgöz-Akdağ, H | Conference | To guide the automotive sectors by analyzing the current situation with the Lean concept and using the necessary technologies. |
| 4 | 2020 | VALUE STREAM MAPPING 4.0: A STRUCTURAL MODELING APPROACH [3] | Rania El Kammouni, Oualid Kamach, Malek Masmoudi | Article | VSM 4.0 for a collaborative work environment for management teams and production process optimization through digitization |
| 5 | | Value Stream Mapping-based Logistics 4.0 Readiness for Thailand Automotive-Part Manufacturers [15] | Boonsothonsatit, G.; Tonchiangsai, K.; & Choowitsakunlert, S. | Article | The article discusses a methodology to transition to logistics 4.0 based on VSM and the adaptation of 4.0 technologies. |
| 6 | | Industry 4.0-based real-time scheduling and dispatching in lean manufacturing systems [16] | Ramadan Muawia Salah Bachir | Article | An Industry 4.0 based Lean framework called DVSM dynamics to digitize Lean manufacturing with Industry 4.0 Lean tools and technologies. |
| 7 | | Lean and Industry 4.0—How to Develop a Lean Digitalization Strategy with the Value Stream Method [17] | Schneider Markus | Article | Methodology that enables manufacturing companies to develop an individual digitalization strategy using Industry 4.0 |
| 8 | 2019 | Industry 4.0: Development of a multi-agent system for dynamic value stream mapping in SMEs [18] | Huang, Z., Kim, J., Sadri, A., Dowey, S., Dargusch | Article | Application of VSM in SMEs is more difficult for complicated product processing and improvements in labor management and facility utilization. |
| 9 | 2018 | Sustainable value stream mapping and technologies of Industry 4.0 in manufacturing process reconfiguration [19] | Phuong, N.A., Guidat, T | Article | RFID, big data and usability improvement as solutions and the impact of their implementation on process sustainability |
| 10 | | Value stream method 4.0: Holistic method to analyze and design value streams in the digital age [8] | Hartmann, L., Meudt, T., Seifermann, S., Metternich, J | Article | Promising new opportunities from digitalization and Industry 4.0 to integrate information flows. |
| 11 | 2017 | From value stream mapping to value stream management—how the static lean method can be further developed to a dynamic management approach using solutions of Industry 4.0 [20] | Lugert, A., Winkler, H. | Article | VSM is common in manufacturing companies and has advantages but also some shortcomings suffered by the current megatrends in the production environment. |
| 12 | | Value stream mapping 4.0: Holistic examination of value stream and information logistics in production [21] | Joachim Metternich | Article | Mapping allows the analysis of process chains and helps to deduce improvement potentials |

**Table 1.** *Cont.*

| N° | Year | Title | Authors | Type | Contenent |
|---|---|---|---|---|---|
| 13 | 2015 | Reduction of Work in Process Inventory and Production Lead Time in a Bearing Industry Using Value Stream Mapping Tool [22] | Rajenthirakumar et Shankar | Article | A case study on the application of VSM in the automotive sector, resulting in a 67% reduction in cycle time by optimizing value-added activities |
| 14 | | Deployment of Lean Management in a packaging workshop and change management [23] | Julia Flauder | Thèse | Lean should extend to all links in the supply chain, upstream and downstream of production, which has long been its focus. |
| 15 | 2014 | Reduction of Wastage Using Value Stream Mapping: Case Study [24] | Rajendra kumar | Article | The application of the VSM on the production flow shows that the non-value-added activities have been optimized such as waiting time, handling time . . . |
| 16 | | Quality Value Stream Mapping [25] | B. Haefner, A. Kraemer, T. Stauss, and G. Lanza | Article | Q-VSM is a tool for the visualization, design and analysis of quality assurance measures for the manufacturing system in the electronics industry |
| 17 | 2012 | The transportation value stream map (TVSM) [26] | Bernardo Villarreal | Article | T-VSM modeling that provides a detailed description of storage and transportation operations |

We find that the new VSM 4.0 method creates a digital and collaborative work environment for management teams and stakeholders in the Industry 4.0 era. It offers logistics production companies a huge benefit in the planning and optimization processes by digitizing the value to make improvements and more accurate decisions.

The objective of our research is to introduce digital and automatic tools to collect data, intelligent means of information processing, data storage medium, and key performance indicators (KPIs) so that the monitoring of the logistics activity and the information management system (IMS) are accurate and consistent and within the framework of sustainable development. The information collected can be visually represented in real-time to logistics managers.

In conclusion, the approach of VSM is very effective and often used in the literature for production with possible extensions to transport, quality, environment, etc. we want to use VSM as a model to support the analysis process of the downstream SC.

According to the literature review, this method is more applied to the automotive industry, especially to the procurement and production process, and very little work has begun to apply it to the downstream supply chain.

## 3. Lean Value Stream Mapping VSM Implementation Method

### 3.1. Value Stream Mapping

Value Stream Mapping was developed by engineer Taiichi Ohno at Toyota to identify and eliminate waste sources [27]. Value Stream Mapping (VSM) is a type of flowchart used to analyze, model, and propose improvements to the processes leading to the delivery of a good or service. It is one of the most frequently used formalisms in companies. Indeed, it is a key element of the Lean methodology to detect and eliminate waste. By grouping all the activities that make up the supply chain, the VSM tool aims to identify non-value-added processes that are considered wasteful. Following this teamwork, improvement solutions are developed to correct these bottlenecks to optimize the production flow [28]. The VSM tool takes into consideration physical flows, but especially information flows [29].

### 3.2. Advantages of the Lean VSM Tool in Logistics

According to our bibliographic study, several authors have shown that the VSM method is an essential tool in Lean Manufacturing and Supply Chains [2,28] for several reasons:

- It helps to go beyond the level of single individual processes and to visualize the entire supply chain as a whole (the VSM map starts with the arrival of the products at the warehouse and ends with the shipment of the products to the final customer);
- Detection of waste and indication of the causes;
- Providing a basis for discussing the importance of the various logistics processes;
- It constitutes a draft for discussion towards a lean approach, the outline of the plan of a future organization;
- The VSM map shows the links between the different product and information flows.

The VSM integrates three components: material flow, information flow, and scheduling. Building a VSM is often the first step in the transformation to lean. The VSM tool identifies problems and provides suggestions on how to transform the process to lean. In other words, the VSM can be used to identify where loss occurs and where value is added. It can also help a company understand what value it is actually providing to its customers [30].

### 3.3. VSM Map Design Process

To analyze the current process of the downstream SC, we have chosen the lean tool of VSM to map the value chain. This tool allows us to analyze all the tasks of the SC with the objective of optimizing the tasks that have a non-added value. This approach offers the possibility of federating human capital around common objectives focused on economic, social, and environmental performance. In this study, we present the development of this VSM project in 6 steps, as shown in Figure 3:

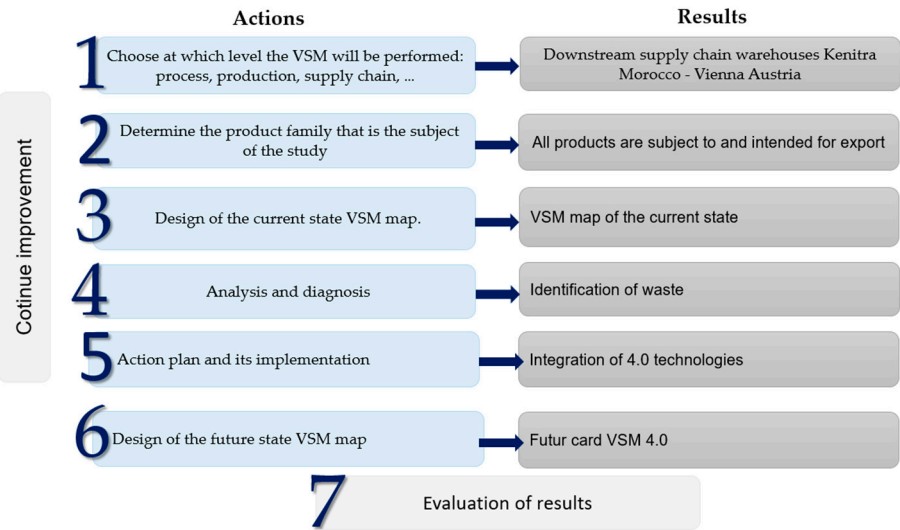

**Figure 3.** VSM design process.

## 4. Contribution of 4.0 Technologies in Logistics

Before approaching the case study of downstream logistics, it is fundamental to conduct a study that establishes the relationship between 4.0 technologies and logistics processes. This work is of great necessity and considered a decision support tool for the implementation of our approach

Table 2 presents a study of the most recent technologies and their impact on the different SC processes in different industries and modern logistics companies.

It is clear that there is a multitude of Industry 4.0 technologies and the disparity of their uses in the industrial world (Table 2). Each of the technologies mentioned can be used according to the needs of the company, the specificities of the sector, and the amount of investment that can be made by the companies. Negotiating the digital shift is imperative for companies today. Knowing the technological building blocks of Industry4.0 will allow decision-makers to ask themselves the right questions about their uses in accordance with the needs and priorities to be defined very early on.

Table 2. The different 4.0 technologies used in the industry.

| | New Technologies | Contribution of the New Technologies on the Logical Processes |
|---|---|---|
| 1 | Big Data | - Comprehensive real-time data collection and evaluation<br>- Rapid data processing<br>- Support decision making<br>- Analyze and separate important data<br>- Support more effective knowledge transfer to achieve business goals [31]. |
| 2 | Data storage in the cloud | - Reduces the initial investment required when implementing a WMS, since instead of acquiring the entire technical infrastructure, a monthly rent is paid for the license.<br>- Large-scale processing<br>- Flexible use for computing and storage [32]. |
| 3 | IOT | - Related to the various sensors that collect information from the physical world and transfer it to digital databases [32].<br>- Improving Supply Chain Visibility<br>- Accurate information in real time [33]. |
| 4 | COBOTS | - Transport of goods<br>- Order picking<br>- Improve supply chain efficiency [31]. |
| 5 | AM<br>(Additive Manufacturing) | - AM refers to a set of automated processes that build products layer by layer, based on three-dimensional models designed in computer-aided design software.<br>- Its objective is to standardize the packaging of the same size [34]. |
| 6 | Drones | - Flying over the logistics sector with the aim of providing support for the performance of certain tasks, such as inventory management [28]. |
| 7 | AI | - Simulate natural intelligence to interpret external data<br>- Learn from this data for descriptive, predictive, or normative analysis in logistics.<br>- Help generate supplies, anticipate customer orders in the warehouse [25]. |
| 8 | AR<br>(Augmented Reality) | - Works with virtual objects that overlap with the existing environment.<br>- Virtual and real information, acquired with a camera, are digitally merged, and represented on a screen<br>- Create an interface between employees and digital products or equipment [10]. |
| 9 | VS<br>(Virtual simulation) | VS is a computer-based modeling system that promotes real-time data to reflect the physical world in a virtual model that includes machines, products, logistics flows and humans [31] |
| 10 | RFID<br>(Radio frequency identification) | - Identify and track without physical contact.<br>- Read, store, and write information on electronic tags using radio waves [35].<br>- Composed of a chip, a tag (antenna role) and a reader.<br>- To be read, the tags need to be present in the radio wave radius of the reader [35]. |
| 11 | Smart Pack | - Connecting packages<br>- Monitor the conditions of transport of goods<br>- Collect time-stamped data on geolocation, shocks, temperature, and humidity<br>- Transmit the data according to the chosen frequency [33]. |
| 12 | Code-barre | - Bar code, or bar code (CAB), is the representation of a numerical or alphanumerical data in the form of a symbol made up of bars and spaces.<br>-The thickness varies according to the symbology used and the data thus coded [36]. |
| 13 | Power BI | - Power BI Warehouse Performance content was created so that warehouse and operations managers can monitor important inbound, outbound and inventory metrics.<br>- It uses warehouse management, product, and other trans-actional data [37]. |

**Table 2.** *Cont.*

| | New Technologies | Contribution of the New Technologies on the Logical Processes |
|---|---|---|
| 14 | WMS | - Guide the receipt and storage of inventory<br>- Optimize order preparation and shipping<br>- Advises on inventory replenishment [38]. |
| 15 | Scan 3D | - A three-dimensional scanner is a device for scanning and 3D acquisition<br>- Analyzes the objects or their close environment to collect precise information on the form and possibly on the appearance (color, texture . . . ) [39].<br>- This device helps us to facilitate the quality control of the goods, 3D modeling. |
| 16 | ERP | - Collects all data from a warehouse into a centralized database<br>- Enable informed and intelligent decision making<br>- By using an ERP system, inventory will be automatically integrated with all aspects of the business, from planning to production to accounting [40]. |
| 17 | ATLS | - Robotic system by which pallets are automatically loaded and unloaded from trucks with little operator intervention.<br>- The ideal solution to speed up and secure the receipt and dispatch of goods in the loading area of the warehouse [41]. |
| 18 | AGV/LGV | - Self-guided vehicles, similar to forklifts, move autonomously and automatically, following a pre-determined or pre-programmed path.<br>- To guide them, two systems are available: self-guided (AGV) or laser guidance (LGV) [25]. |
| 19 | Stacker cranes | Automation of product entry and exit operations.<br>Elimination of errors derived from manual management.<br>Control and updating of inventory management [42]. |

Table 2, presents a study of the most recent technologies used in different industries and modern logistics companies.

We note that 4.0 technologies clearly contribute to the improvement of the supply chain and provide efficient solutions for forecasting the flow of supplies, sales, distribution, and location of products in the warehouse.

## 5. Case Study of an International Supply Chain in the Automotive Industry

### 5.1. Presentation of the Study

Our case study was carried out in a company of Automotive Equipment Manufacturers located in Morocco (EAKM) to deliver to its warehouse in Vienna (EAVA) and distribute products throughout Europe. Its official headquarters is in Austria. The group specializes in designing and realizing electrical cables for the automotive sector, whose production is done in Kenitra, Morocco. The production of electrical and electronic components is entirely intended for export and, more particularly, for the large car manufacturers in Europe.

The circuit through which the product passes is three stages and presented on Figure 4:

(1) Road transport from Kenitra to Tangier by truck;
(2) Sea transport from Tangier to Algeciras by a commercial ship;
(3) Road transport from Algeciras to Vienna by the same truck;
(4) It should be noted that for this transport, the goods remain in the truck from its loading in Kenitra until its unloading in Vienna, unlike the first method of transport which was the goods in containers.

Our work focuses on the downstream supply chain of the distribution process that transports the finished product to the company's warehouse in Vienna, Austria.

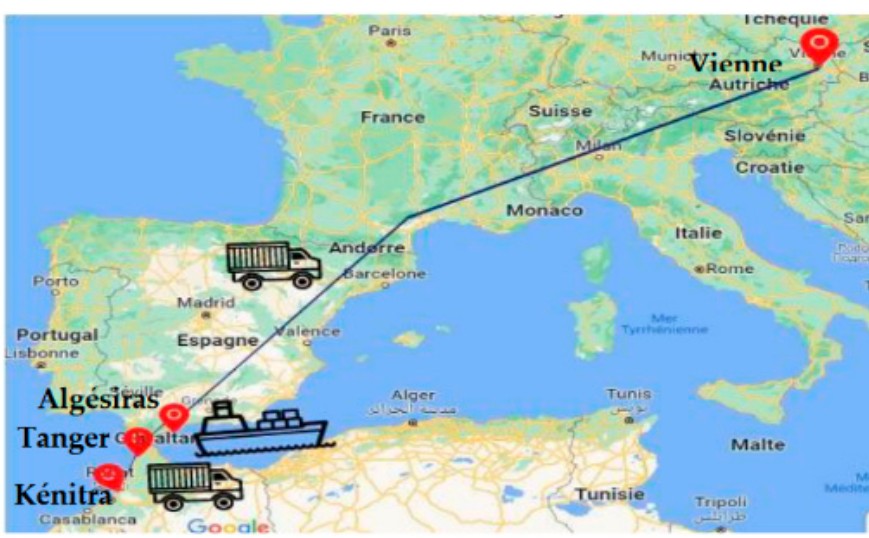

**Figure 4.** Road and sea transport route.

*5.2. Analysis of the Current State*

To analyze the current state of our value stream mapping, we will proceed through three steps:

- Identify the different steps in our supply chain, from order preparation to delivery to customers (see Table 3).
- Consider the logistics competitiveness of each country involved in this supply chain by proposing a new process.
- Present the map of the current state of the process. Each task is presented by a box attached with a data box that contains information such as the number of operators in each workstation with the cycle time of all tasks, Lead Time, Down Time, and Process Times.

**Table 3.** The tasks of the distribution process.

| Activities | Definition |
|---|---|
| Product pickup | The accumulation of the stock of finished products requires several rounds of collection during the rotation of all the teams. |
| Grouping of packages | By grouping, we mean putting the products in racks in the shipping area. |
| Sorting of Finished Product (FP)/Customer | The dispatch area is divided into different zones to store one batch per customer. |
| Sorting PF/Reference | In the shipping area, sorting can also be done by reference. |
| Palletizing | La mise en palette de produit finis commence dès qu'une référence atteint la quantité exigée par le client. |
| Transfer of packages | The palletization of finished products starts as soon as a reference reaches the quantity required by the customer. The pallets having the required standards of packaging and quantity will have to be transferred by means of the radio frequency gun from warehouse F to warehouse E |
| Edition of the expedition file | Preparation of the export file (Invoices . . . ) |
| Packaging and shipping | The pallets transferred to the warehouse and well checked are now ready to be stretch-wrapped with the help of handling means. |
| Moving of finished pallets | The finished pallets must be loaded respecting the priority to the pallets that must be the first to be unloaded. |
| Loading of packages | The filling of the trailer begins each time there are finished pallets while respecting the loading norms in force. |
| Road Transport Kenitra-Tangier | Transport the goods to the port of Tangier Med. |
| Sea Transport Tangier-Algeciras | Checking in Tangier Med and boarding for Algeciras. |
| Road transport Algeria-Vienna | Transporting the goods from the port of Algeciras to the warehouse in Vienna |
| Reception and unloading | Unloading and storage of the finished products in the EAVA warehouse in Austria. |

The different steps of the downstream process in the company are identified and projected on the VSM map. The objective is to visualize in a clearer way the positions that require the most attention, for example time management, number of operators, cycle time, and non-value-added time (see Figure 5). Following this first mapping of the current state, the company wants to improve its competitiveness. In this framework, we will propose an improvement by introducing new technologies of industry 4.0, whose objective is to optimize this downstream supply chain. Therefore, we will first propose the most used 4.0 technologies in modern companies and the most suitable technologies for each workstation; this leads us to obtain a new future state mapping called VSM 4.0 (future map), which will receive new data and information from the whole SC, rationalizing both the management of information flows and product flows.

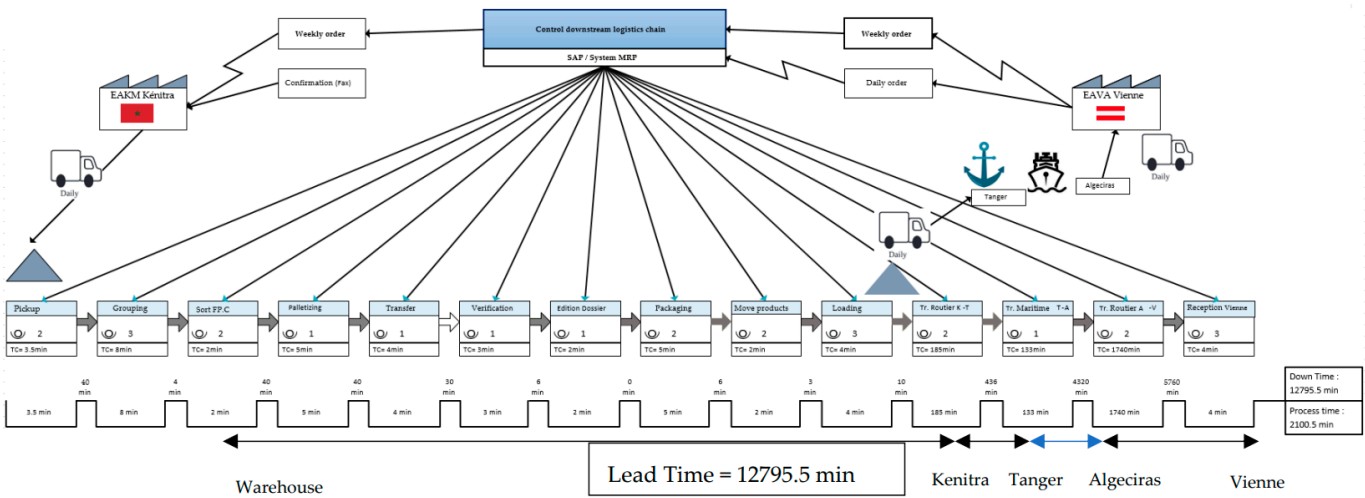

**Figure 5.** Mapping of the current state.

*5.3. Implementation of the Methodology*

In order to implement the research work, a methodology of implementation in 4.0 was implemented. It was divided into three phases:

First, according to the previous approach Figure 3, the first phase is to study and analyze the existing process in order to obtain the objectives, identify the process, and understand the industrial context of the company, then determine the sources of waste in the process and finally propose solutions.

The second phase consists in implementing 4.0 technologies in the supply chain. Study the compatibility of these technologies with each process and establish a VSM 4.0 mapping, where each process will be connected to this mapping (Future Map).

Finally, phase III corresponds to the collection and processing of data by the WMS, ERP, and intermediate communication tools and their connectivity. We present in Figure 6 below the results of the methodology implementation.

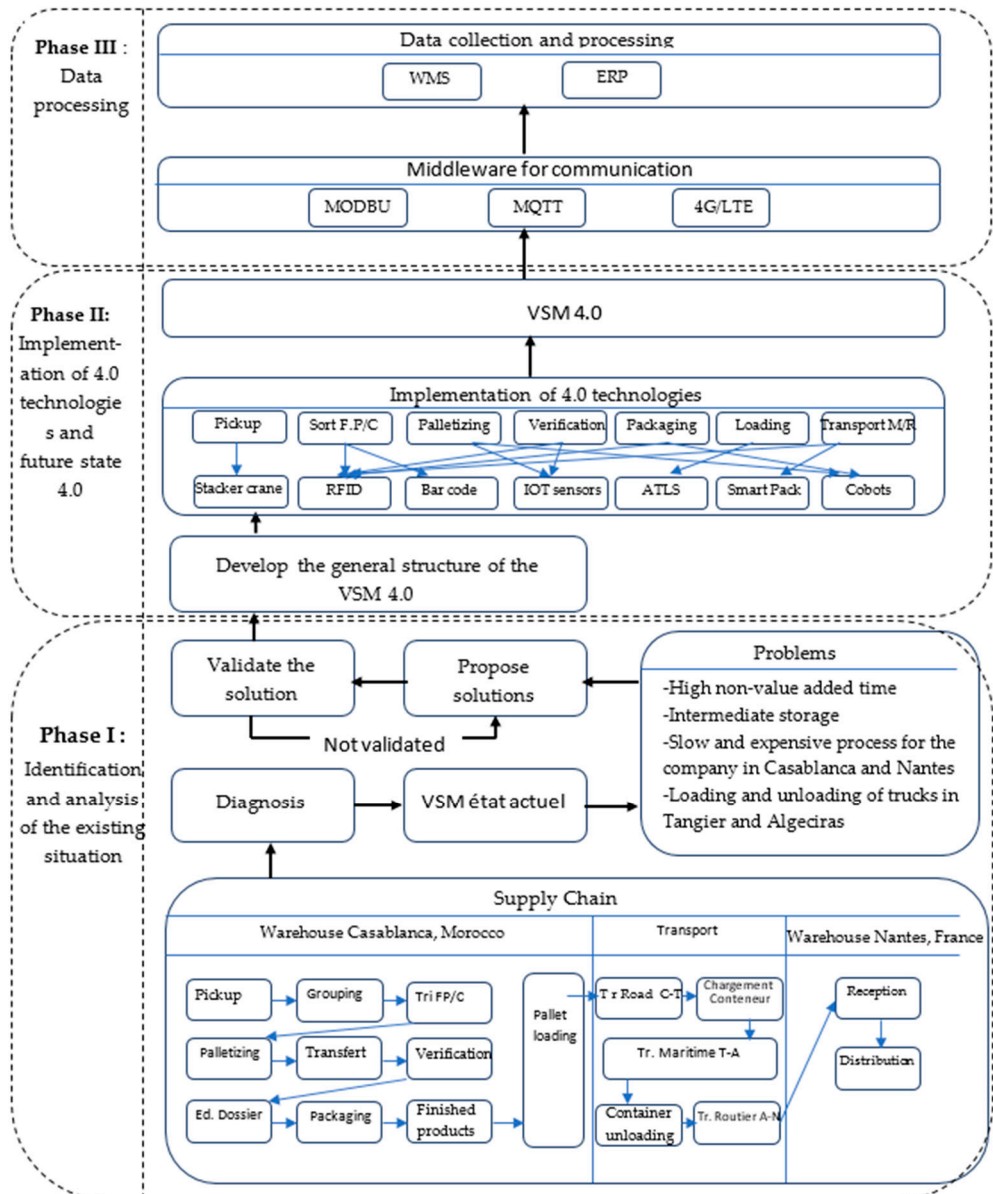

**Figure 6.** VSM 4.0 implementation methodology.

## 5.4. Implementation of Technologies

We have established the relationship between the 4.0 technologies and the different tasks of the downstream logistics process (see Table 4). The company has selected the technologies according to the following criteria:

- The most efficient way to optimize time and gain more cost.
- The investment cost is not too high for the implementation of these technologies.
- The training and mastery of these technologies by its employees.
- The transition time between the current state and the future state is not long.

**Table 4.** Assignment and adaptability of technologies to process functions.

| Functions / Technologies | Big Data | Cloud Computing | IoT | Cobots | AM | Drones | AI | AR | VS | RFID | Smart Pack | Bar Code | Power BI | WMS | Scan-3D | ERP | ATLS | AGV/LGV | Transtockeur |
|---|---|---|---|---|---|---|---|---|---|---|---|---|---|---|---|---|---|---|---|
| Pick-up & grouping of products | | | | X | | | | X | | X | | | | X | | X | | | X |
| Product sorting | X | X | | X | X | | X | X | | X | | X | X | | X | | | | |
| Palletization & transfer of packages | | | X | X | | | | | | X | | | | | | | | X | X |
| Editing the file | X | X | | | | | X | X | | X | | | X | X | | X | | | |
| Packaging & Shipping | | | | X | | | X | X | | X | | | | | | X | | | |
| Moving finished pallets | X | X | | | | | | | | X | | | | | | | | X | |
| Loading and unloading | | | | X | X | | | | | | | | | X | | | X | | X |
| Road, sea and warehouse delivery | X | X | | | | | | | | X | X | | | | | | | | |

The 4.0 technologies chosen by the company are implemented in the different functions of the logistic process. A training of the personnel took place for a mastery of these technologies to accompany this digital transformation. After six months of work the results were very satisfactory (see Table 5).

**Table 5.** Contribution of technologies for each process.

| Tasks | Contribution of New Technologies |
|---|---|
| Pickup & consolidation | **Stacker cranes 1, and Augmented Reality**<br>- Product location and identification in the warehouse<br>- Products are scanned and registered in real time,<br>- Transport of products from production according to traceability to the grouping area<br>- Optimize the role of the operator and optimize space<br>- Merging of picking and grouping<br>- The storage capacity is doubled<br>- The circuit of movements is optimized<br>- Reduced number of accidents<br>- Reduced energy consumption<br>- Reduced number of damaged packages<br>**WMS**<br>- Receipt of goods from imported files, the characteristics of goods received and purchase orders (order number, reference, quantities, etc.) are recorded in a file<br>- Direct connection between the warehouse and the company's ERP system, which has led to better order management and a significant reduction in the delivery time to the customer<br>- Possibility to follow each product inside the warehouse by controlling the inventory. |
| Sorting products | **Bar codes**<br>- Automatic sorting where you already know where each product is and to which category it belongs<br>- Reduce the verification time for the operator |
| Palletizing & transfer | **AGV and augmented reality**<br>- Optimize the role of the operators in this stage by automatic palletizing system<br>- Reduce largely the time of palletization in this stage<br>- Reduced number of accidents<br>- Eeduced energy consumption |

**Table 5.** *Cont.*

| Tasks | Contribution of New Technologies |
|---|---|
| Verification and editing of the file | **RFID & Scan**<br>- Instantly detect which product is which and fill in the file data automatically<br>- The shipping warehouse manager here will have the role of confirming the exit of the product without the need for manual entry.<br>- It will save time for the final verification of the products.<br>- Follow-up of the product until the loading in the trucks. |
| Packing and shipping | **RFID**<br>- In this step, RFID technology allows us to track each finished product after wrapping and labeling, the WMS software allows us to control the goods before shipment and their allocation to the truck, we can add a third operator to speed up the operation. |
| Moving of finished pallets | **AGV**<br>- For fast and organized movement without the need for operator intervention<br>- Reduction of movements inside the warehouse, and facilitates the work of the operators<br>- The movement circuit is optimized<br>- Reduced number of accidents<br>- Reduced energy consumption |
| Charging | **ATLS**<br>- Reduction of the loading time in the trucks<br>- Rare intervention of the operator whose role will be to supervise<br>- All the boxes are of standardized size which will allow to gain in storage space and to optimize the distances between boxes<br>- Optimize the company's logistic resources . . . |
| Road and sea transport and warehouse delivery | **RFID & Smart pack**<br>- Traceability and tracking of products<br>- Optimize the time of search and transport of boxes<br>- Locate where each product is at any time |
| Distribution | **AGV and stacker crane**<br>- Gain in reactivity and delivery time<br>- The circuit of movements is optimized<br>- Reduced number of accidents<br>- Reduced energy consumption<br>- Reduced number of damaged packages |

### 5.5. Future State Mapping VSM 4.0

Figure 7 shows the future state 4.0 mapping of the downstream supply chain. Our value stream mapping is optimized with fewer tasks. The total number of workstations within the company is reduced to 9. In summary, after implementing this new mapping along the supply chain we obtained the following improvement results by implementing VSM 4.0 within the international supply chain:

- Automation and fusion of several processes: handling, transport, data collection, etc.
- The use of 4.0 technologies offers a fast product flow, and where data can be transmitted in real time.
- The VSM will continuously receive new data and information from the entire supply chain, allowing both the management of flows and the identification of waste that hinders the competitiveness of the company.
- Track the product throughout the process to identify and locate it quickly and above all to eliminate the possibility of loss. Simultaneous updating facilitates the consolidation of data, making it easier to make decisions.
- Change and improvement in the transportation of goods.

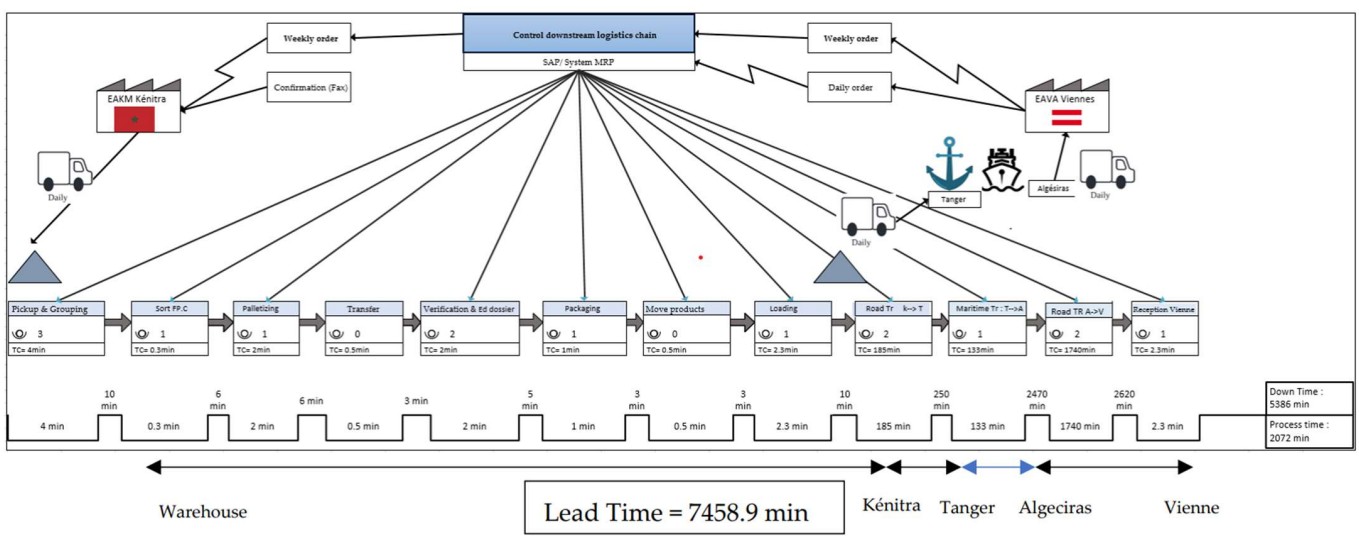

**Figure 7.** Future State 4.0 Mapping.

## 6. Analysis of the Results

### 6.1. Implementation of the Technologies (Future State) in the SC

After implementing the previously proposed solutions, Table 6 presents the various improvements on the downstream part of the logistic chain from Kenitra-Morocco to Vienna-Austria. We merged the collection and grouping using stacker cranes and the verification with the edition of the file. This action allowed us to optimize the process cycle time, the information flow in the company, as well as the role of the operators. By using new 4.0 technologies, automation, and connectivity, we have managed to optimize these processes reducing time, risks, and energy consumption.

**Table 6.** Comparison table between current state and future state 4.0 in warehouses.

| Process | Current State | | | Improvement through New Technologies | Future State 4.0 (min) | | |
|---|---|---|---|---|---|---|---|
| | Operators | Cycle Time (min) | Non-Value Added Time (min) | | Operators | Temps de Cycle (min) | Temps de Non-Valeur Ajoutée (min) |
| Pickup | 2 | 3,5 | 40 | Stacker cranes, RA | 3 | 4 | 10 |
| Grouping | 3 | 8 | 4 | | | | |
| Sort FP/C | 2 | 2 | 40 | Bar codes | 1 | 0,3 | 6 |
| Palletizing | 1 | 5 | 40 | AGV | 1 | 2 | 6 |
| Transfer | 1 | 4 | 30 | AGV | 0 | 0,5 | 3 |
| Verification | 1 | 3 | 6 | RFID & Scan | 2 | 2 | 5 |
| Edition Dossier | 1 | 2 | 0 | | | | |
| Packaging | 2 | 5 | 6 | AGV | 1 | 1 | 3 |
| Move products | 2 | 2 | 3 | AGV | 0 | 0,5 | 3 |
| Loading EAKM | 3 | 4 | 10 | Stacker cranes, ATLS, AGV, RA | 1 | 2,3 | 10 |
| EAVA unloading and distribution | 3 | 4 | | | 1 | 2,3 | |
| Total | 21 | 42,5 | 179 | | 10 | 14,9 | 46 |

### 6.2. SC Performance after Integration of New Technologies

Currently, an SC's design must consider the interests of all integrated actors. Logistics is part of a "sustainable" movement, and Sustainable Development (SD) is becoming

increasingly important within companies. According to this vision, companies are obliged to think about the extended economic, environmental, and social/societal performance. In this context, we present the improvement of the company's performance after the integration of new technologies.

6.2.1. Economic Performance

We have seen a strong benefit in time saving such as:

- Time and cost savings in Tangier and Algeciras: loading and unloading avoided;
- Reduction of loading and unloading time by using ATLS tools.

Table 7, we compare some indicators after the implementation of this new card, and we note the improvements and efficiency introduced.

**Table 7.** Gain between current state and future state 4.0.

| Comparison Table | | | |
|---|---|---|---|
| **Indicator** | **Current State** | **Future State 4.0** | **% Gain** |
| Number of positions (tasks) | 11 | 9 | 20% |
| Lead Time | 8 days 21 h | 5 days 4 h | 41% |
| Number of operators | 21 | 10 | 52% |

These results are obtained thanks to the technologies introduced in this supply chain Table 6. For transportation, the value-added time and the non-value-added time is presented in Table 8.

**Table 8.** Transportation Comparison Table between Current State and Future State 4.0.

| Process | Process Time (min) | Non-Value-Added Time (Min) | |
|---|---|---|---|
| | | **Current State** | **Future State 4.0** |
| Road Transport Kenitra-Tanger | 185 | 436 | 250 |
| Sea Transport Tangier-Algeciras | 133 | 4320 | 2470 |
| Road Transport Algeciras-Vienna | 1740 | 5760 | 2620 |
| Total Time | 2058 | 10,516 | 5340 |

Finally, the Lead Time and the time between the order and final delivery of the product to the customer are compared.

This section may be divided by subheadings. It should provide a concise and precise description of the experimental results, their interpretation, and the experimental conclusions that can be drawn.

In order to analyze the current state of our value stream mapping, we will proceed by:

- Identifying the different steps in our supply chain, from order preparation to delivery to customers.
- Taking into account the logistics competitiveness of each country involved in this supply chain by proposing a new process.
- Presenting a map of the current state of the process. Each task is presented by a box attached with a data box that contains information such as the number of operators in each workstation with the cycle time of all tasks, Lead Time, Down Time, and Process Times. (see Figure 8).

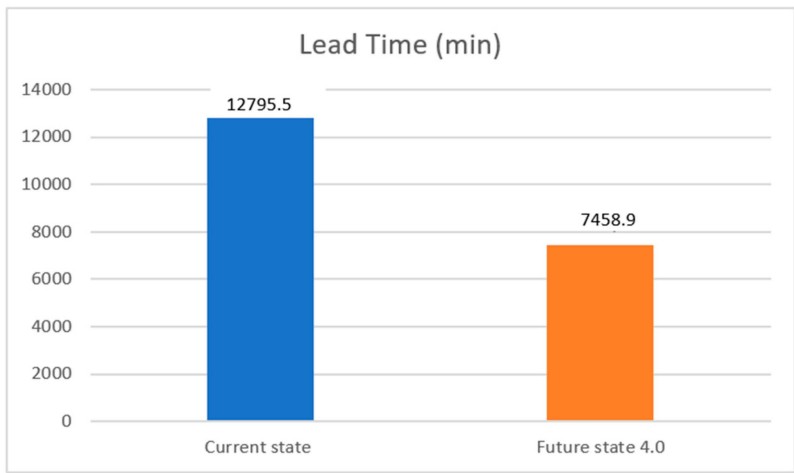

**Figure 8.** Comparison between Time to Value Added current and future state.

With the integration of these technologies, we witness a performance in time, distance travelled, truck fill rate, control of the distance travelled, and stop management.

The reactivity and flexibility led to improved logistics services, customer satisfaction, and increased turnover.

### 6.2.2. Social Performance

With the introduction of the Lean VSM method combined with 4.0 technologies, automated handling, and the motivation and development of the staff through the number of trainings, the number of conferences/seminars on sustainability, and the level of effort and communication to understand all stakeholders of the company and the concept of sustainability, it has been noted for a duration of the year 2021:

A space-saving in the warehouse of 20% is due to the use of modern handling equipment such as stacker cranes. The new organization and the space-saving have significantly reduced the travel circuit in the warehouse. The use of augmented reality has facilitated the location and inventory of products. The movement of products is reduced by the AGVs and the number of staff too. All these technologies had a very positive impact on the logistics staff. We have identified three indicators to show this impact on the social environment of the company as presented in Table 9.

**Table 9.** Social indicators.

| Indicators | The Year 2020 | The Year 2021 |
|---|---|---|
| The number of accidents | - 25 accidents<br>- 23 minors less than 10 days sick leave<br>- 2 major more than two months of sick leave | 5 minors |
| Absenteeism rate | 63 | 9 |
| the number of hours of maintenance intervention | 242 | 27 |

We noticed that the number of accidents and the rate of absenteeism was very low after one year. With the introduction of a preventive maintenance of the material, the number of hours of intervention on the machines was much reduced.

As a result of these changes, we noticed a climate of trust, team spirit, motivation, employee satisfaction, and a collective intelligence that developed.

### 6.2.3. Environmental Performance

We noticed a positive impact on the management of resources in particular:

- Reduction in the cost of energy consumption that can be explained by the switch to an electric automated handling that replaced the traditional fuel-based handling means;
- Elimination of noise (noise pollution) and $CO_2$ emission;
- Reduction of waste (hydrocarbon waste, sludge, grease, used oil);
- The optimization of the movement circuit at the warehouse level has eliminated the risk of damage to packages during movement and consequently reduced the number of claims by customers for obsolete products.

## 7. Discussion

There was a gap of 5336.6 min, equivalent to three days and some hours in Lead Time. This was the result of optimization of the supply chain by modifications to the transport process and implementation of new 4.0 technologies in the warehouse of Kenitra and Vienna.

First of all, in EAKM Kenitra-Morocco, by using 4.0 technologies such as automatic handling equipment: stacker cranes, AGVs, and ATLS, we facilitated the role of the operators, and, as a result, these tasks will no longer be carried out manually but by machines and more quickly than before. This means that the number of operators in the process can be optimized, especially in the receiving area of the EAVA warehouse in Austria.

Secondly, through connectivity technologies, such as augmented reality, RFID, and Smart pack, connected to the WMS of the warehouse and the ERP of the company to follow the quantity, location, and state of the product at every moment in the warehouses of Kenitra and Vienna and the transport are tracked.

For the existing means of transport, the container was transported by a truck to Tangier, unloaded at the port, and loaded on a commercial ship. Then, in Algeciras, again, it is loaded in a truck which transports the goods to Vienna.

The proposed solution is to transport the goods and store them in a truck that makes the whole trip from EAVA to Vienna. This will avoid the costs of loading and unloading operations in both ports, which are very expensive.

Finally, at the transport level, we add a second transport staff to reduce the non-value-added time at this level for the journey to Europe.

## 8. Conclusions

The objective of our study was to provide a methodology based on Value Stream Mapping 4.0 for the continuous improvement of the downstream SC distribution process.

We established a VSM map to analyze the current state, then the design of the future state with optimization of some tasks considered competitive for the country of origin and the integration of 4.0 technologies accepted by the company from the point of view of implementation time and budget. Through this work, we were able to improve economic, social, and environmental performance. With the Lean Value Stream Mapping 4.0 tool, all products and information flow in a value chain from suppliers to customers are analyzed and optimized.

The integration of 4.0 technologies makes the VSM tool capable of monitoring value streams in real time to resolve potential waste quickly. This new intelligent organization provides a fully integrated logistics environment where data is transmitted in real time between warehouses in the countries concerned and the customers. The results obtained are encouraging and have an impact on economic and social performance. The introduction of new technologies in a company requires a significant investment cost but is quickly paid back by the improvements offered.

**Author Contributions:** Conceptualization; methodology, software, formal analysis, investigation, data curation, writing—original draft preparation, Y.E.K.; writing—review and editing, A.E.K. and S.E. All authors have read and agreed to the published version of the manuscript.

**Funding:** This research received no external funding.

**Acknowledgments:** We would like to thank the EAKM distribution team for allowing us to collaborate with our research team by providing us with the downstream logistics process data. We would like to thank the engineers and managers of the logistics distribution chain for the time they devoted to us, and for agreeing to implement the methodology developed within their international distribution chain.

**Conflicts of Interest:** The authors declare no conflict of interest.

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
