# Peer review of "Optimization of the Sustainable Distribution Supply Chain Using the Lean Value Stream Mapping 4.0 Tool: A Case Study of the Automotive Wiring Industry"

_processes, doi:10.3390/pr10091671_

Round 1

Reviewer 1 Report

I am very grateful to have been able to participate in this review.

The abstract and introduction are sufficiently clear, and largely convey the content of the article.

However, in the literature review, which is very well structured, I miss the passage from SCOPUS to WoS and the introduction and citation of well-valued articles in the area such as http://doi.org/10.3233/JIFS-179639 .

The statement of the problem is more than detailed and well explained, as well as the development of the study and the conclusions.

It is true that the conclusions could and should be somewhat more specific and pragmatic.

For the rest, the article has merit in its approach, its development, execution, and layout.

Author Response

Hello,

I thank you for your pertinent remarks.

We have taken all these remarks into account and you will find the text on the revised article.

Sincerely

Reviewer 2 Report

- line 71 - check your English spelling in the title

- check your paper formatting -part of the title is bold, and part is not.

- why are some titles in Table 1 written in CAPS lock and others are not - standardize and again check your spelling

- also literature review is not putting 3 graphs and 1 table mentioning how many papers have been written in the last 5 years! The literature review is something completely else

- line 131 - who developed lean tool of VSM - reference is missing

- line 138 - source for the figure 3?

- line 156 - title for table 2 is missing

- format table 2 and make it standardized - so every row has the same format

- for some sentences (definitions), references (sources) are missing

- In the theoretical part of the paper, you have put everything, but this is still too weak for the scientific paper - you need to make the literature review stronger, and this is not made by just putting several papers on the topic. You need to show the connection to your empirical part; I can't see it now. You need to significantly improve this part of the paper

- if you are presenting the table in line 190, put title Table 3

- in line 232, you state Table 5, but where is table 4?

- in the line 305 you are referring to the table 7 - is this a correct table?

- again, calling a wrong table number in line 307 - did you check your table numbers and which one you are calling in the text?

- table 9 and Figure 7 are showing same thing - choose one

- size of the fonts are different in part  7.1.2.

- what is the connection of part 7.1.2. to the research topic - how did you get the mentioned results, which methodology was used, ......

- some sentences in the part Discussion need to be rewritten in order to be understood - i.e. The chosen means of transport is trucks instead of hundred year olds

- check the literature list since I don't see that it fits to the journal template.

Your paper is interesting, but it is missing a lot and needs to be significantly improved.

-

Author Response

Hello,

Thank you for your pertinent remarks which we have taken into account.

Attached you will find the answers.

Sincerely

Round 2

Reviewer 2 Report

Dear authors,

Improvement can be seen in the paper.

I would like to suggest checking the grammar and typing errors - for instance, in line 85 - "contribute to it. . In addition" - there is a double ..

Also - line 70 -  Revue is what? Review?

Why did you choose only the Scopus database and not WoS database - this should be explained in the methodology.

Why do you have Table 8 and Figure 7 - they show the same

In 6.1.1. Economic performance - you are putting some items not proven with numbers - i.e. customer satisfaction -when did you measure it, and how? You measured (presented) only the lead time - the rest is based on your assumptions (no numbers)

Also, you state the second driver - what is his influence on the costs? Although the lead time is shortened, we have a second driver - increased cost - or?

In 6.1.2. - again, font size

In 6.1.2 -  don't you think 20% more space in the warehouse is more economical than social benefit and preventive maintenance as well?

How you measured the improvements you mention in Social performance - there is nothing about this in your paper - you can not state that there is an improvement unless you can prove it - so the whole social performance part needs to be proven with numbers and methodology.

The way you have listed the references is not following the journal template, so you need to correct that as well - this was mentioned in the first review and you didn't correct it.

Author Response

Hello,
Thank you for your feedback.
You will find attached the answers to your remarks. 
Sincerely
